# Coupling Coordination Relationship between Tourism Industry and Ecological Civilization: A Case Study of Guangdong Province in China

**Dan Yuan [1],\* and Guanwei Jang [2]**

1 School of Management, Guangdong University of Science and Technology, Dongguan 523000, China
2 School of Business, Shaoguan University, Shaoguan 512005, China
\* Correspondence: yuandan@gdust.edu.cn; Tel.: +86-135-5665-6648

**Abstract:** Ecological civilization has a significant role in the sustainable development of the regional tourism industry. The high dependence of tourism on ecological civilization contributes to the complex interaction between the two. How to coordinate the relationship between tourism and ecological civilization has affected the sustainable development and ecological civilization strategy of Guangdong Province and even the whole of China. Previous quantitative research on the coupling and coordination of the two has certain limitations. One limitation is the lack of dynamic prediction. In this paper, an evaluation index system of the relationship between tourism and ecological civilization is constructed. Quantitative methodologies such as entropy-weight, gray relation analysis, and obstacle degree are integrated. Meanwhile, the gray model (GM), GM (1,1) model is used for prediction. The data sample is based on the years 2005 to 2021 in Guangdong Province, China. The main results are as follows: (1) Tourism and ecological civilization both rise slowly from a low starting point. Affected by the epidemic, tourism declines rapidly in 2020 and rises slowly in 2021, but it still lags behind ecological civilization. Both continue a slow upward trend over the next decade. (2) The degree of coupling coordination has undergone a fluctuating evolution from moderate disorder to mild coordination and falls back to the barely coordination stage from 2020 to 2021. In the next decade, the coupling coordination between the two will be gradually optimized. The coupling coordination degree will be greater than 0.8 having entered a benign coordination stage in 2028; (3) The gray relation and obstacle factors show different dynamics. High gray relation comes from the ecological civilization. Partial factors are still in the region with high gray relation and high obstacle degree. The results of this paper are expected to contribute to a more appropriate ecotourism model and provide some enlightenment for implementing the effectively coordinated sustainable development of tourism and ecological civilization in Guangdong Province, China, as well as those areas with the same industrial characteristics.

**Keywords:** tourism industry; ecological civilization; sustainable development; gray relational analysis; obstacle degree

## 1. Introduction

Ecological civilization is a millennium plan closely related to sustainable development and has been an action toward environment protection in China. It is also an important starting point for high-quality development under the new normal of economic development. How to accelerate the realization of harmony and unity between ecological civilization and high-quality economic development has always been a research hot spot. Meanwhile, ecological civilization is also necessary in order to realize "beautiful China", which has been written into numerous national development strategies in China [1]. As a hot spot of social investment and a strategic pillar industry of the national economy, the tourism industry always plays an important role in promoting ecological civilization. It is clearly pointed out in the 13th Five-Year Plan for Tourism Development in China that tourism, with its

endogenously innovation-leading, coordinated, open, interactive, environment-friendly and co-sharing characteristics, is highly compatible with the five development concepts. It is also one of the most advantageous industries in the construction of ecological civilization and should actively become an advantageous industry under the new normal.

In China, the number of A-level scenic spots has been increased from 6042 to 14,332, and the number of 5A scenic spots has increased from 144 to 306, with a significant increase in the proportion of medium and high-level scenic spots from 2012 to 2021. The comprehensive benefits of tourism are being significantly improved again, which has become an important driving force for consumption and growth and has been of great significance in the urbanization process [2]. However, the tourism industry may be a double-edged sword, especially the extensive development mode dominated by the traditional tourism resources can easily cause a series of ecological problems, such as resource waste, scarcity, over development, and environmental damage [3,4]. Meanwhile, tourism is also increasingly dependent on resources, and there are environmental considerations [5]. Therefore, in the process of tourism headway, how to solve the dilemma between the extensive tourism economic development and ecological environment [6] and realize the coordinated development between tourism and ecological civilization [7] has become critical.

Tourism and ecological civilization mutually promote and restrict. As pointed out in some studies in the existing literature, tourism and ecological civilization, tourism urbanization and ecological civilization, as well as tourism–regional economy–ecological civilization are highly coupled. A coupling coordination model can accurately reflect the degree of coordinated development and mutual influence among these systems. However, these existing studies mainly concentrate on the interaction feedback mechanism, the measurement of coordination effect, and the status analysis of the coupling relationship. There are some shortages regarding the coupling and coordination relationship prediction between regional tourism and ecological civilization [8].

The main contributions of this paper are as follows: (1) From the perspective of time scale, the coupling coordination, gray relation and obstacle degree between tourism and ecological civilization are explored, which is a good attempt compared with the previous mono-status analysis of coupling coordination degree; (2) The GM (1,1) prediction model of gray system theory is added to the coupling coordination degree model. Based on the sample data from 2005 to 2021, the tourism, ecological civilization and their coupling coordination degree are predicted and analyzed, which can contribute to understanding the development tendency and making effective decisions for the future development; (3) The comprehensive evaluation of gray correlation degree and obstacle factors is sorted out according to the four-quadrant method. Factors that affect the coordinated development of tourism and ecological civilization are divided into four quadrants: low-gray relation low- obstacle, low-gray relation high-obstacle, high-gray relation low-obstacle, and high-gray relation high-obstacle. Then main influencing factors are refined, which can be more conducive to the targeted practical applications and promote the efficient and coordinated development between tourism and ecological civilization.

## 2. Literature Review

### 2.1. Tourism Industry Evaluation

Along with the application of sustainable and synergistic theory in tourism research, focusing on various tourism systems is becoming more and more popular, and the relationship between symbiotic interaction and coupled development among various systems has been recognized to a large extent. Kurniawan et al. studied the complexity of the sustainable development of tourism in a small island in combination with social ecosystem methods and pointed out that the development of tourism will be strongly affected by the natural environment, and the development of tourism itself will also affect the natural environment as well [9]. Streimikiene et al. analyzed the significance of tourism sustainable development from the enhancing competitiveness perspective, and they believed that institutional, social, cultural, technological, environmental, economic, and other factors can

promote the high-quality development of tourism [10]. Some related studies are shown in Table 1.

**Table 1.** Some recent studies that evaluate the tourism industry.

| Researcher(s) | Methodology | Description |
|---|---|---|
| [11] | Coupling coordination model and division of characteristic level | Considering four criteria of tourism industry scale, tourism industry structure, factor investment in tourism industry, and tourism industry development efficiency to evaluate the tourism industry growth system. |
| [12] | Delphi technique and analytical hierarchy method | Refined four main criteria and 33 sub-indicators to measure the sustainable rural tourism development. |
| [13] | Delphi and fuzzy analytic hierarchy method | Proposed a low-carbon tourism development evaluation model consisting of 6 aspects and 16 evaluation categories with 53 indicators. |
| [14] | Fuzzy analytic hierarchy process method | Extracted 16 influence factors to empirically analyze the determinants of tourism attractiveness. |
| [15] | Artificial intelligence neural network model | Put forward a tourism resources evaluation system containing 5 primary indicators and 12 secondary indicators. |
| [16] | Information entropy weight and gray system prediction model | Considering two criteria of tourism performance and tourism scale, used seven sub-indicators to measure tourism system. |
| [17] | Entropy and spatial autocorrelation methods and coupling coordination degree | Used 10 indicators to measure the tourism industry, such as revenue from domestic tourism, inbound tourism income and so on. |
| [18] | Interactive coercion model, coupling coordination degree model and gravity model | Summarized four criteria of population, economy, society, and space to assess the tourism urbanization. |

### 2.2. Ecological Civilization Evaluation

The first "ecological civilization construction" national strategy was developed by the Communist Party of China in 2012. It aims to keep the homeostasis of ecology, environment, natural resource, and economic development [19]. At present, it has become prominent in the contribution to global sustainable development [20]. Research on the quality of ecological civilization construction is also becoming abundant. The research dimensions include land spatial layout, resource utilization, ecological economy, ecological agriculture, and ecological industry [21]. The research scales cover the whole country, provinces, and urban agglomerations. The research methods are mostly evaluated by building evaluation index systems. However, there are still inconsistencies in the academic understanding of ecological civilization. Some relevant studies on ecological civilization evaluation are shown in Table 2.

**Table 2.** Some recent studies about evaluation of ecological civilization.

| Researcher(s) | Methodology | Description |
|---|---|---|
| [18] | Interactive coercion model, coupling coordination degree model and gravity model | The pressure-state-response (PSR) model is feasible to explain the eco-environment. |
| [20] | Comprehensive assessments | Ecological system is a complex circulation, including environment, energy flow, and information exchange, and a synthetical weight-based index of ecological civilization assessment with 33 indicators is proposed. |
| [22] | Data envelopment analysis (DEA) | Selected five perspectives of social, economic, political, cultural, and ecological to analyze the ecological civilization performance in China. |
| [23] | Exponential variable weight (EVW) method, TOPSIS model and obstacle degree model | Proposed an extended PSR index system of urban ecological civilization from economy, resources, environment, and society aspects. |
| [24] | Linear correlation | Used three indicators of ecosystem services, ecological footprint, and GDP per capita to measure the regional ecological civilization. |
| [25] | AHP and gray correlation analysis and coupling model | Developed four criteria of ecological economy, ecological environment, ecological habitat, and ecological mechanism to evaluate ecological civilization. |
| [26] | Cloud model and coupling coordination degree model | Proposed a provincial-level evaluation system of ecological civilization level including ecological economy, ecological society, ecological nature. |
| [27] | Factor analysis | Explored an index system including living and working in peace, resources and environment, and development potential. |

The dialectical relationship between tourism and ecological civilization has been explored from the perspective of economics and ecology. Some relevant studies are shown in Table 3.

In summary, research on the evaluation of tourism and ecological civilization and the relationship between the two has gradually deepened, and the methods are becoming diversified as well. A more standardized research framework of "evaluation-influencing factors (mechanism)" has also formed. The application of a coupling coordination model is still universal. For the methodology, the empirical orientation is more obvious, and the study area is differential involving several countries, the whole country, the industrial belt, the province, or a single city.

**Table 3.** A review of the literature on the relationship between tourism and ecological civilization.

| Researcher(s) | Study Area | Description |
| --- | --- | --- |
| [2] | 30 provinces in China | Tourism development is evaluated along with the energy and environment consumption, showing obvious "Kuznets curve" phenomena in the long term. |
| [8] | China | Coordination between tourism and ecological for the next five years is predicted, and it takes a long time to realize the coordinated development. |
| [17] | 12 provinces in Western China | Coupling and coordination between tourism and environment is complex with spatial correlation and dependence characteristic. |
| [18] | 12 provinces in Western China | Tourism brings complex ecological effects, moving toward an inverted U-shaped curve and dynamic development of coupling and coordination. |
| [28] | Liupanshui, China | Coupling coordination of civilization and tourism showed an overall upward, the tourism relatively lags. |
| [29] | Nagasaki Prefecture, Japan | The combination of coupling coordination degree model and pressure-state-response can help better solve complex coupling relationships and formulate sustainable strategies for tourism and the ecological environment. |
| [30] | 31 countries, African | There exists cointegration and a non-linear relationship between tourism and environment degradation. |
| [31] | France, USA, France, Italy, China | Empirically analyzes how tourism may promote environmentally sustainable development, and China must more effectively implement sustainable tourism environmental development policies to reduce environmental pollution caused by tourism. |

However, there is not much on future prediction, especially lacking dynamic information and development trend prediction analysis [32]. Thus, it is worth considering the practical research on discipline integration in early warning and regulation, the improvement of big data application, the construction of a tourism and ecological synergy mechanism, and the strengthening of the tourism and ecological synergy mechanism [33]. Prediction of the coupling and coordination between the two will facilitate putting forward targeted coupling and coordination measures and providing a more scientific reference for achieving sustainable development and optimizing regulation [8]. In this regard, this paper has predicted the future evolution of coupling and coordination of tourism and ecological civilization in Guangdong Province based on a GM (1,1) model.

### 2.3. Coupling Mechanism between Tourism and Ecological Civilization

Coupling coordination is a process of mutual coordination and supplementation between two systems with differences that have much common, such as the mutual relationships between tourism, urbanization, ecological environment, and economic development. The tourism industry (TI) is a kind of comprehensive consumption, including a series of elements and economic activities such as "food, housing, transportation, entertainment, tourism and shopping". Along with the transformation of the consumption concept,

tourism has a depth of connotation, as it is not only a new form of industry but integrated into a way of life, learning, and growth. Ecological civilization (EC) reflects industrial civilization and a social and technological imagination, which combines certain cultural, moral values, and political goals [21]. Tourism and ecological civilization are two complex and interrelated systems, which have been consistent in goals and values and also have natural coupling characteristics [28], as shown in Figure 1.

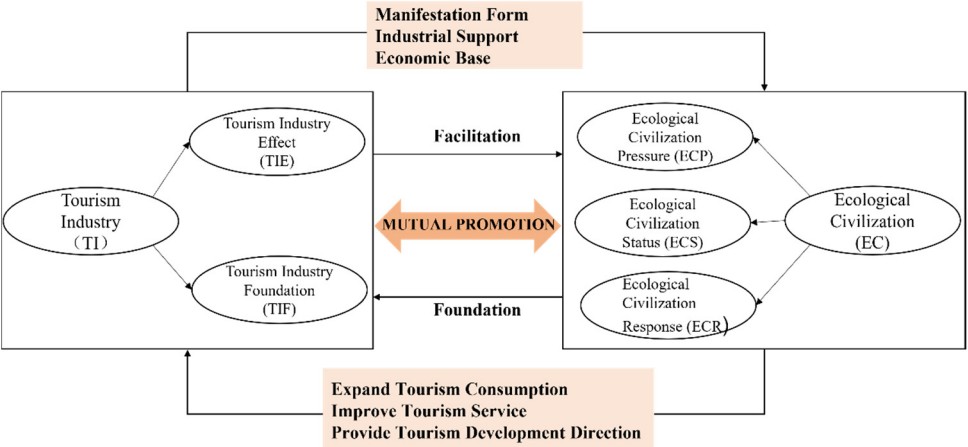

**Figure 1.** Coupling and coordination mechanism of TI and EC.

On the one hand, the development of tourism has boosted ecological civilization. Firstly, as a typical resource-saving and environment-friendly industry, tourism has obvious comprehensive characteristics, which can promote the development of relevant industries as well as improve the cultural conservation and living standards of the masses. Tourism helps to effectively achieve material ecological civilization, cultural ecological civilization, and social ecological civilization. These are the main manifestation of regional ecological civilization achievements. Secondly, tourism is characterized by strong comprehensiveness, great relevance, and an obvious multiplier effect [34]. The development of tourism is beneficial for other industries upgrading and structure adjustment. These can provide good industrial support for the construction of ecological civilization in a direct and indirect way. Thirdly, tourism is a typical "people friendly" and "people rich" industry that creates employment and solves employment problems. In this respect, tourism can provide a better economic basis for ecological civilization. For example, Zhangjiajie, a new tourism city in China, has experienced ecological overload due to early over-development and over-utilization; as a result, tourism has stagnated. However, the formulation of a series of low-carbon tourism policies and incentives has effectively avoided ecological risks. At present, the happiness perception of the residents is very high, the economic income has greatly improved, and its ecological environment has steadily reached the national secondary standard for four consecutive years. Zhangjiajie has been rated as the "most ecologically competitive city in China".

On the other hand, ecological civilization is the foundation for the sustainable development of tourism, because ecological civilization provides rich resources and environmental elements for tourism and maintains the sustainable development of tourism destinations, which is the objective basis for the orderly development of tourism [35]. Firstly, a beautiful environment and good ecology are the most basic and main stimulus to the public's tourism demand, which can effectively expand tourism consumption and increase the attractiveness of tourism destinations [36]. In the rating of tourist attractions, ecological environment and ecological health near scenic spots account for major proportions. Secondly, ecological civilization can not only optimize the tourism environment and improve the quality of tourism but also enable tourists to enjoy more during the travel and effectively improve the quality of tourism services. Thirdly, with the constant upgrading of consumption, various industries are undergoing optimization strategies and industrial restructuring. Tourism is

also being affected without exception. Nowadays, eco-tourism, supported by a beautiful environment and good ecological environment, has been a key to the transformation and upgrading of tourism and sustainable development. Eco-tourism has become a mainstream, deriving a variety of sustainable tourism modes and a series of eco-tourism products. Thus, ecological civilization will become a primary consideration for the sustainable development of tourism since it points out the direction for the further development of tourism.

For the tourism industry subsystem, the "Tourism Industry Effect (TIE)" measures the economic benefits brought by the tourism industry, and it reflects the development scale of the tourism industry, such as tourism foreign exchange earnings, domestic tourism earnings, inbound, domestic tourists, etc. The "Tourism Industry Foundation (TIF)" measures some resources of tourism industry development, reflecting the current situation. It is calculated by the number of hotels and restaurants, the number of A-level and above tourist attractions, employment in accommodation and catering, and highway mileage.

The ecological civilization subsystem is based on the causal model of "P (pressure)–S (state)–R (response)" [37]. The pressure subsystem of ecological civilization (ECP) reflects the threat of some industrial production activities to the surrounding environment and natural resources, with a negative effect such as industrial wastewater and exhaust emissions. The subsystem of ecological civilization status (ECS) reflects the stability of the ecosystem in the region and is closely related to the natural and healthy vegetation conditions in the region [38], such as forest coverage rate, green coverage rate in built-up area, etc. The ecological civilization response (ECR) measures the collection of some behaviors that individuals and society deal with in relation to environment improvement. It is the logical starting point and end point of the whole ecological civilization system.

As mentioned above, tourism has promoted the construction of ecological civilization, and it also relies on the ecological civilization. Meanwhile, ecological civilization provides a good environmental foundation for tourism development. Some elements of ecological civilization can restrict tourism development in some way. The orderly and coordinated development of the two can bring more linkage effects without doubt, which also has turned into a focus of regional sustainable development.

## 3. Study Areas and Data Source

### 3.1. Study Area

Guangdong Province, referred to as Yue, is in the south region of China, between 20°13′ N to 25°31′ N and 109°39′ E to 117°19′ E. It is adjected to Hong Kong, Macao, Guangxi, Hunan, Jiangxi, and Fujian (Figure 2). The terrain is generally high in the north and low in the south, with mountains and high hills in the north and plains and platforms in the south, with an average elevation of more than 70 to 120 km. It is one of the regions with the richest light, heat and water resources in China. The total area of Guangdong Province is 179,700 km$^2$, and the resident population is 126.84 million as of the end of 2021, which is an increase of 600,000 over the end of the previous year. The annual GDP in 2021 is 124,369.67 billion RMB, which is an increase of 8.0% over the previous year. The urbanization rate of the province's permanent population has increased from 66.2% in 2010 to 74.2% in 2020; the per capita urban public green area has increased from 13.29 square meters in 2010 to 18.13 square meters in 2020.

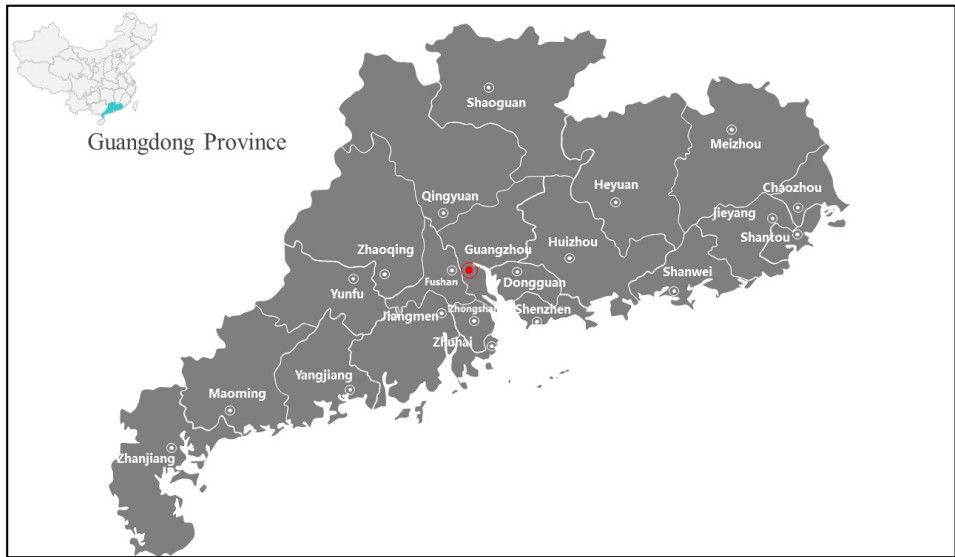

**Figure 2.** Geographical location of the study area.

*3.2. Data Source*

The data considered in this paper are mainly from the Statistical Yearbook of Guangdong Province from 2005 to 2021, and some tourism-related data are from the Tourism Yearbook of Guangdong Province and China Tourism Yearbook of the same period. The data on the ecological environment are from the China Environmental Statistical Yearbook and the Statistical Bulletin of National Economic and Social Development of Guangdong Province in the same period. In addition, some of the missing indicators are selected from the Guangdong Provincial Department of Natural Resources, the Provincial Department of Ecology and Environment and the Provincial Department of Culture and Tourism, and these are calculated by linear interpolation.

**4. Methods**

As mentioned, different systematic evaluation systems related to the tourism industry and ecological civilization have existed in the literature review. Research methods are diversified as well. The application of coupling theory is very common. Nevertheless, merely taking a single angle of the tourism industry or ecological civilization and describing the status of coupling and coordinated development between systems may not be a useful indicator for decision making. It is necessary to effectively integrate the current situation and prediction in the evaluation and analysis. Therefore, this paper focuses on forecasting and finding out the influencing factors of the tourism industry and ecological civilization evaluation system in order to better support some management decisions.

Specifically, this paper takes Guangdong Province as the research object to analyze the comprehensive development between the tourism and ecological civilization (TI-EC) systems. Firstly, the range method is used to normalize the initial data of the index system without dimension, which is convenient for comprehensive evaluation of the index. Secondly, the entropy method is used to assign weight to each secondary index, which avoids the influence of subjective factors and makes the evaluation method more objective and accurate. Finally, the coupling coordination, GM (1,1), gray relation analysis (GRA), and obstacle degree are used to effectively reflect the interaction and synergistic relationship between TI and EC systems under different temporal conditions. A flowchart of the methodology used in this paper is shown in Figure 3.

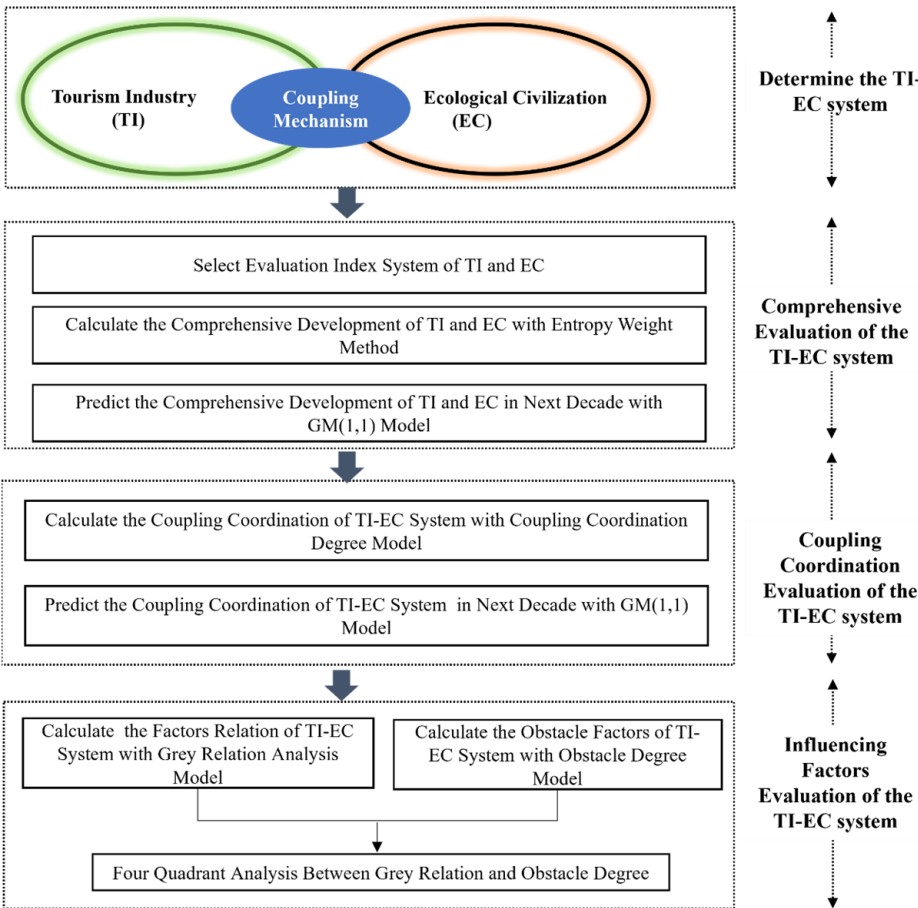

**Figure 3.** Flowchart of the used methodology.

*4.1. The Evaluation Index System*

The evaluation index system is the basis of exploring the coupling and coordination between the tourism industry and ecological civilization. However, the preliminary indicator system is only a framework based on qualitative methods, which may be subjective. Thus, the selected index system should represent both the respective system and reflect the close relationship between the two systems. Thus, this paper sorted out the literature related to tourism and the ecological civilization system from the principle of a coupling coordination system. In addition, this paper also follows the relevant studies to pay attention to the essence of the tourism industry and ecological civilization system, and it comprehensively considers the actual development of Guangdong Province to adhere to the systematic, hierarchical, representative, practicability, comprehensiveness, and availability principles of index selection [39]. Then, twenty specific sub-indicators including the tourism effect, tourism foundation, ecological civilization pressure, ecological civilization status, and ecological civilization response are used to interpret the comprehensive development level of the tourism industry and ecological civilization construction system in Guangdong Province (shown in Table 4).

**Table 4.** Evaluation index system of TI-EC system.

| System Level | Target Layer | Index Layer | Index Unit | KMO and Cumulative Contribution Rate |
|---|---|---|---|---|
| Tourism Industry System (TI) | Tourism Industry Effect (TIE) | Foreign exchange earnings T1 (+) | 100,000,000 yuan | 0.707 (72.047%) |
| | | Domestic tourism earnings T2 (+) | 100,000,000 yuan | |
| | | Inbound tourists T3 (+) | 10,000 people | |
| | | Domestic tourists T4 (+) | 10,000 people | |
| | | Ratio of tourism revenue to GDP T5 (+) | % | |
| | | Ratio of tourism revenue to the tertiary industry T6 (+) | % | |
| | Tourism Industry Foundation (TIF) | Number of hotels T7 (+) | / | 0.808 (88.56%) |
| | | A-level and above tourist attractions T8 (+) | / | |
| | | Employment in accommodation and catering T9 (+) | 10,000 people | |
| | | Highway mileage T10 (+) | km | |
| Ecological Civilization System (EC) | Ecological Civilization Pressure (ECP) | Industrial waste gas emissions E1 (-) | 100,000,000 m³ | 0.784 (92.55%) |
| | | Industrial SO₂ emissions E2 (-) | 10,000 tons | |
| | | Industrial solid wastes produced E3 (-) | 10,000 tons | |
| | | Industrial wastewater E4 (-) | 100,000,000 tons | |
| | Ecological Civilization Status (ECS) | Forest coverage rate E5 (+) | % | 0.695 (90.782%) |
| | | Green coverage rate in built-up area E6 (+) | % | |
| | | Per capita public green area E7 (+) | m² | |
| | Ecological Civilization Response (ECR) | Rate of sewage centralized disposal E8 (+) | % | 0.597 (76.481%) |
| | | Rate of garbage harmless disposal E9 (+) | % | |
| | | Comprehensive utilization rate of industrial solid waste E10 (+) | % | |

### 4.2. Entropy Weight Method

The entropy weight method is an objective weighting method, reflecting the importance of indicators with the strength and non-uniformity of data itself. This paper establishes the initial data matrix $X = (X_{ij})_{m \times n}$. To minimize the magnitude influence, the data were standardized. Assuming that $X_{ij}$ is the original data of the $j$th index in the $i$th year, $X'_{ij}$ is the standardized post-processing data, $X_{jmin}$ is the minimum value of the $j$th index, and $X_{jmax}$ is the maximum value of the $j$th index. These are calculated as follows:

$$X'_{ij} = \begin{cases} \frac{X_{ij} - X_{jmin}}{X_{jmax} - X_{jmin}} + 0.01, & X_{ij} \text{ is positive} \\ \frac{X_{jmax} - X_{ij}}{X_{jmax} - X_{jmin}} + 0.01, & X_{ij} \text{ is negative} \end{cases} \tag{1}$$

The entropy weight method is used to assign the value of each index. Firstly, we calculate the proportion $Y_{ij}$ of the $i$th evaluation object in the $j$th index, which is shown in Formula (2). Secondly, we calculate $E_j$ and the difference coefficient $D_j$ in the information entropy of the $j$th term index, as shown in Formulas (3) and (4). Thirdly, we calculate the weight of the $j$th index $W_j$, which is shown in Formula (5). Finally, we calculate the comprehensive evaluation value of the $i$th evaluation object $U_i$, which is shown in Formula (6).

$$Y_{ij} = \frac{x'_{ij}}{\sum_{i=1}^{m} x'_{ij}} \tag{2}$$

$$E_j = -\frac{1}{\ln(m)} \times \sum_{i=1}^{m} Y_{ij} \ln Y_{ij} \tag{3}$$

$$D_j = 1 - E_j \tag{4}$$

$$W_j = \frac{D_j}{\sum_{j=1}^{n} D_j} \tag{5}$$

$$U_i = \sum_{j=1}^{n} X'_{ij} W_j \tag{6}$$

### 4.3. Gray Model GM (1,1)

Gray system theory is a scientific theory based on weakening the randomness of a data series to find a potential law and then establishing a gray model for prediction according to some solution. One of its advantages is its ability to use small samples and incomplete information for prediction [40]. The GM (1,1) model is the first-order differential equation model of a single variable of gray theory. The model includes a one-time accumulation of original data to generate new sequence data, a smoothness test, construction of GM (1,1) differential equation, model solution and model verification.

For complete details, parameters, and calculation formulas of the GM (1,1) model, the study of Liu and Yang can be consulted in detail [41]. The accuracy criteria for the GM (1,1) model prediction [8] are shown in Table 5.

**Table 5.** Accuracy criteria of GM (1,1) model prediction.

| Accuracy Criteria | C Value | $p$ Value |
|---|---|---|
| Excellent | $C \leq 0.35$ | $p \geq 0.95$ |
| Good | $0.35 < C \leq 0.5$ | $0.8 \leq p < 0.95$ |
| Qualified | $0.5 < C \leq 0.65$ | $0.7 \leq p < 0.8$ |
| Failed | $C > 0.65$ | $p < 0.7$ |

### 4.4. Coupling Coordination Degree Model

The coupling degree measures the degree of interaction between systems but does not indicate the level of system coordination. In this regard, this paper refers to the relevant studies [42], also adopting coupling degree C (Formula (7)), comprehensive development index T (Formula (8)) and coupling coordination degree D (Formula (9)) to comprehensively evaluate the coordination between the tourism industry and ecological civilization.

$$C = \sqrt{(U_{TI} \times U_{EC})/(U_{TI} + U_{EC})^2} \tag{7}$$

$$T = \alpha \times U_{TI} + \beta \times U_{EC} \tag{8}$$

$$D = \sqrt{C \times T} \tag{9}$$

$U_{TI}$ and $U_{EC}$ are the comprehensive evaluation values of the tourism industry system and ecological civilization system, respectively, $\alpha$ and $\beta$ are the undetermined coefficients, and the sum is 1. Considering the mutual integration of the TI-EC system, and as mentioned in the majority of the previous studies, this paper selects $\alpha = \beta = 0.5$ as well [43]. Moreover, the coupling degree C and coupling coordination degree D are both located in the [0, 1] interval. The greater value of D is, the higher level of sub-systems coupling coordination [44], that is, the higher level of coupling coordination development between tourism and ecological civilization. The specific values and classification of coupling coordination degree D are shown in Table 6.

**Table 6.** Classification standard of coupling coordination degree.

| Coordination Interval | Coordination Level | Coordination Interval | Coordination Level |
|---|---|---|---|
| 0~0.2 | Extreme Disorder | 0.6~0.7 | Mild Coordination |
| 0.2~0.4 | Moderate Disorder | 0.7~0.8 | Moderate Coordination |
| 0.4~0.5 | Mild Disorder | 0.8~0.9 | Benign Coordination |
| 0.5~0.6 | Barely Coordination | 0.9~1.0 | High-Quality Coordination |

*4.5. Gray Relation Analysis (GRA)*

Gray relational analysis (GRA) is a core part of gray system theory. It is used for the system analysis of incomplete information description, and it is also called gray correlational analysis or gray incidence analysis. Now, gray relational analysis has been well used in probabilistic linguistic environments, forecasting expert weights, and dealing with MCDP problems and DHMADM processes [45]. The gray relational analysis model is used to measure the degree of correlation between factors according to the similarity or difference of the change trend between factors. If the degree of synchronization between factors is the same, it means that the correlation degree is high; otherwise, the correlation degree is low. The specific calculation steps are as follows:

1.  Calculate the K-point gray relational coefficient $\xi_i(k)$.

$$\xi_i(k) = \frac{\min\limits_{i}\min\limits_{k}|x_0(k) - x_i(k)| + \rho\max\limits_{i}\max\limits_{k}|x_0(k) - x_i(k)|}{|x_0(k) - x_i(k)| + \rho\max\limits_{i}\max\limits_{k}|x_0(k) - x_i(k)|} \tag{10}$$

$x_0(k)$ is the reference sequence, $x_i(k)$ is the alternative sequence, $k$ is the sequence length of the series, $\rho$ is the distinguishing coefficient, the value range is generally [0, 1], and 0.5 is generally appropriate [46].

2.  Calculate the gray correlation degree $R_i$.

The average value of K-point gray relational coefficient $\xi_i(k)$ is used as the gray correlation degree between the reference and alternative sequence.

$$R_i = \frac{1}{n}\sum_{i=1}^{n}\xi_i(k) \tag{11}$$

*4.6. Obstacle Degree Model*

Based on the degree of coupling and coordination between the TI-EC system, this paper introduces an obstacle degree model to analyze the main obstacle factors, namely as following:

$$O = \frac{(1 - X'_{ij}) \times W_j \times 100\%}{\sum[1 - (1 - X'_{ij}) \times W_j]} \tag{12}$$

In addition, the higher the O value, the greater the negative impact of the indicator on the sub-system [47]. It also means these indicators need more attention.

## 5. Results and Discussion

*5.1. Analysis of Comprehensive Development*

According to Formulas (1)–(6), the comprehensive values $U_{TI}$ and $U_{EC}$ of the TI-EC system in Guangdong Province from 2005 to 2021 are obtained, and the development evolution is also drawn accordingly (Figure 4).

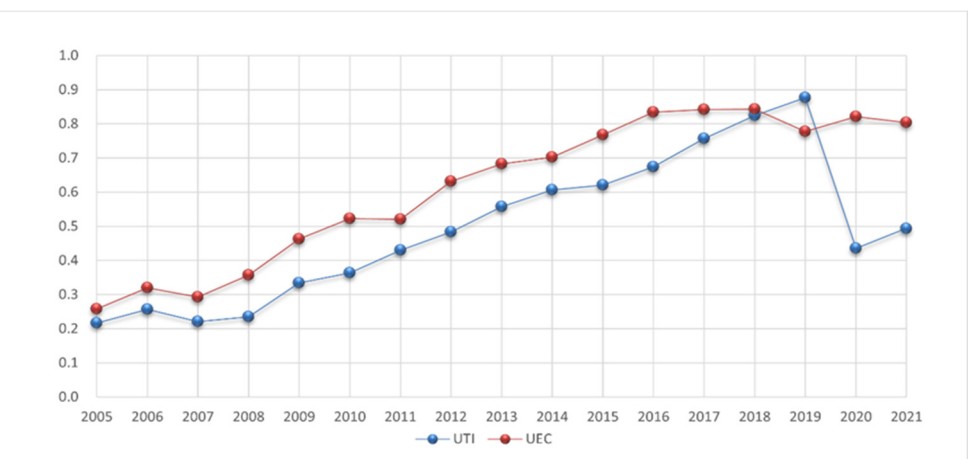

**Figure 4.** TI-EC system comprehensive development in Guangdong Province (Year: 2005~2021).

From the comprehensive development, the tourism industry in Guangdong Province shows an upward trend from 2005 to 2021. However, due to the epidemic, there is a significant decline in 2020, and an increase trend begins in 2021. However, it does not completely recover to the previous trend. The average comprehensive benefit value of the tourism industry development index is 0.49, with an average annual growth rate of 5.27%. On the time scale, the overall development of tourism is lower than that of ecological civilization. Before 2009, the development of the tourism industry in Guangdong Province is relatively slow, and the growth rate increases after 2009. The reason is that since the State Council issued the *Opinions on Accelerating the Development of Tourism* in 2010, Guangdong Province issued a series of documents such as *Tourism Regulations of Guangdong Province*, *Decision on Accelerating the Reform and Development of Tourism and Building A Strong Tourism Province*, and *Implementation Opinions on Promoting the Integrated Development of Culture and Tourism*, which clearly proposed to build into a national tourism comprehensive reform a demonstration zone with high international standards, radiating south China, serving the whole country, influencing the Asia-Pacific region and facing the world. In recent years, Guangdong Province has once again proposed to take high-quality development as the goal, integrated development as the main line, and reform and innovation as the driving force to accelerate the development of culture and tourism into important support for the construction of spiritual civilization and material civilization. All these aims have promoted tourism development to a new level and have accelerated tourism development in Guangdong Province. Therefore, the tourism of Guangdong Province has shown a trend of a low starting point but continuous upward development as shown in the above figure. If there is no epidemic, it is likely to exceed the ecological civilization in 2020.

In addition, on the time scale, from 2005 to 2021, the average comprehensive value of ecological civilization was 0.61, and the average annual growth rate is 7.36% in Guangdong Province. Ecological civilization has fluctuated to a certain extent, showing an overall upward trend. However, there is a downward trend from 2018 to 2019. Further review of the data shows that the industrial solid wastes produced increased significantly in this period. It is related to the late start of ecological civilization in Guangdong Province as pointed out in some literature [48]. The 18th CPC National Congress clearly put forward the importance of ecological civilization construction in relation to people's well-being and the long-term future of the nation. In recent years, Guangdong Province has indeed launched a series of important measures related to ecological environment protection and governance, which has significantly improved the quality of the ecological environment and the development index of ecological civilization construction. However, some long-term accumulated ecological and environmental problems are concentrated, such as the degradation of ecosystem function, prominent environmental pollution problems, extensive production mode, and imperfect ecological civilization system, which still bring severe challenges to ecological civilization in Guangdong Province.

Furthermore, the comprehensive development of tourism in the next decade is predicted based on the GM (1,1) model. It indicates a relatively stable upward trend, as shown in Figure 5.

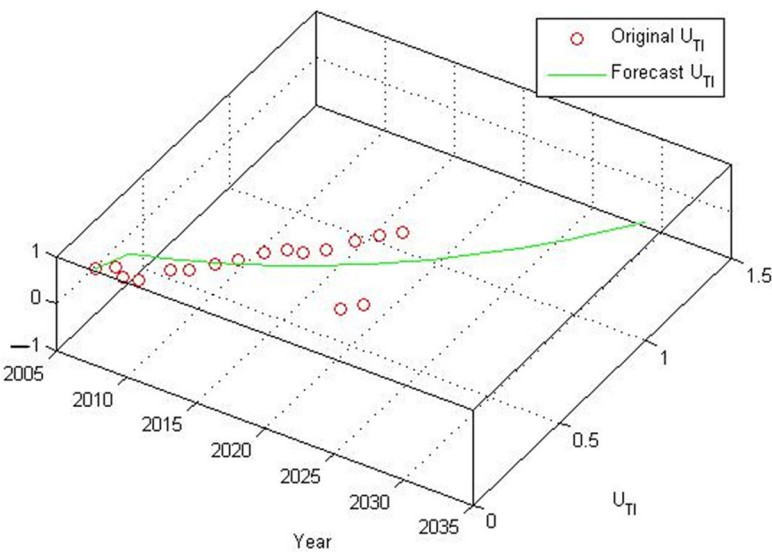

**Figure 5.** GM (1,1) prediction analysis of $U_{TI}$ in the next decade.

Using the same method, the prediction of the comprehensive development of tourism in the next decade is obtained as well. It also shows a stable upward trend, as shown in Figure 6.

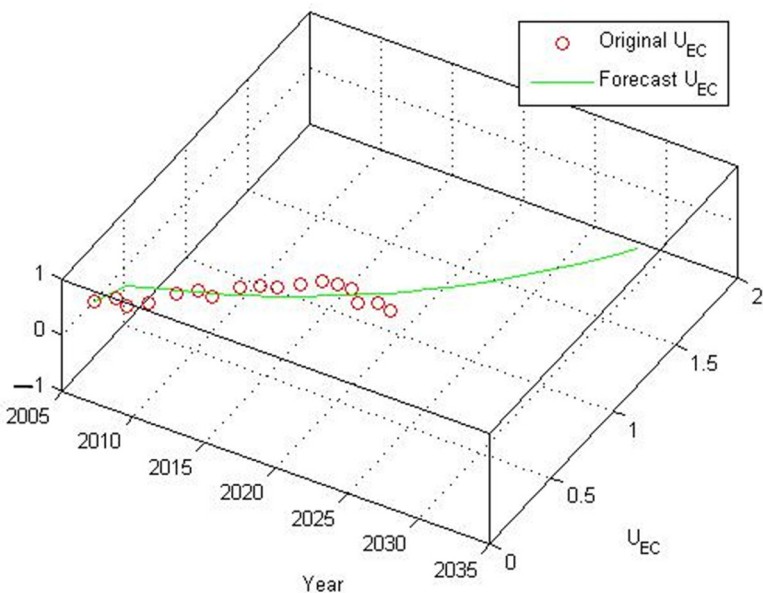

**Figure 6.** GM (1,1) prediction analysis of $U_{EC}$ in the next decade.

### 5.2. Analysis of Coupling Coordination Characteristics

According to Formulas (7)–(9), the coupling degree *C*, comprehensive development *T*, coupling coordination degree *D*, and coordination stage are shown as in Table 7.

**Table 7.** Coordinated development indicators of the TI-EC system (2005–2021).

| Year | Coupling Degree Index C | Comprehensive Development Index T | Coupling Coordination Degree Index D | Coordination Stage |
|------|------|------|------|------|
| 2005 | 0.248 | 0.238 | 0.344 | Moderate Disorder |
| 2006 | 0.247 | 0.288 | 0.379 | Moderate Disorder |
| 2007 | 0.245 | 0.257 | 0.357 | Moderate Disorder |
| 2008 | 0.239 | 0.296 | 0.381 | Moderate Disorder |
| 2009 | 0.243 | 0.398 | 0.443 | Mild Disorder |
| 2010 | 0.242 | 0.443 | 0.467 | Mild Disorder |
| 2011 | 0.248 | 0.475 | 0.486 | Mild Disorder |
| 2012 | 0.246 | 0.557 | 0.526 | Barely Coordination |
| 2013 | 0.247 | 0.620 | 0.555 | Barely Coordination |
| 2014 | 0.249 | 0.655 | 0.571 | Barely Coordination |
| 2015 | 0.247 | 0.694 | 0.588 | Barely Coordination |
| 2016 | 0.247 | 0.754 | 0.612 | Mild Coordination |
| 2017 | 0.249 | 0.800 | 0.632 | Mild Coordination |
| 2018 | 0.250 | 0.833 | 0.645 | Mild Coordination |
| 2019 | 0.249 | 0.827 | 0.642 | Mild Coordination |
| 2020 | 0.227 | 0.628 | 0.547 | Barely Coordination |
| 2021 | 0.236 | 0.649 | 0.561 | Barely Coordination |
| Average | 0.245 | 0.554 | 0.514 | Barely Coordination |

Table 7 shows that the coupling coordination degree between tourism and ecological civilization in Guangdong Province is not high, and it has an average value is 0.514, which is within the stage of barely coordination. Specifically, there is a moderate disorder state from 2005 to 2008. During this period, the tourism industry and ecological civilization construction in Guangdong Province did not receive much attention, and the interaction between the two was weak. Then, the trend transforms into gradual improvement from 2009 to 2021. The coupling and coordination degree gradually improve from a mild moderate disorder to mild coordination. Although there exists an obvious decline in 2020 due to the impact of the epidemic and other environmental factors, a recovery trend appears in 2021. This may be closely related to the gradual adjustment of industrial structure in Guangdong Province, the improvement of ecological and cultural infrastructure, and the transformation and development of low input, low consumption, pollution, and high efficiency [49]. However, during this period, the tourism and ecological civilization have not yet reached the high-quality coordination stage, which indicates that there is still much room for improvement for the TI-EC system development.

Furthermore, using Matlab2019 software to apply the GM (1,1) model to the prediction, this paper draws a conclusion that the gray prediction model is sufficient to analyze the coupling and coordination between the tourism industry and ecological civilization in Guangdong Province, which is shown in Formula (13).

$$Y_D = 13.0375e^{0.0311(t-2005)} - 12.6935, \text{ t is the year.} \tag{13}$$

The gray prediction of tourism and ecological civilization is also obtained, as shown in Table 8.

**Table 8.** GM (1,1) prediction of the TI-EC system coupling coordination (2005–2031).

| Year | Coupling Coordination Degree Index D | Model Fitting Value D′ | Relative Error | Analogue Coordination Stage |
|---|---|---|---|---|
| 2005 | 0.344 | 0.344 | 0.000 | Moderate Disorder |
| 2006 | 0.379 | 0.412 | 0.087 | Mild Disorder |
| 2007 | 0.357 | 0.425 | 0.190 | Mild Disorder |
| 2008 | 0.381 | 0.438 | 0.150 | Mild Disorder |
| 2009 | 0.443 | 0.452 | 0.021 | Mild Disorder |
| 2010 | 0.467 | 0.466 | −0.001 | Mild Disorder |
| 2011 | 0.486 | 0.481 | −0.010 | Mild Disorder |
| 2012 | 0.526 | 0.496 | −0.056 | Mild Disorder |
| 2013 | 0.555 | 0.512 | −0.077 | Barely Coordination |
| 2014 | 0.571 | 0.528 | −0.075 | Barely Coordination |
| 2015 | 0.588 | 0.545 | −0.073 | Barely Coordination |
| 2016 | 0.612 | 0.562 | −0.081 | Barely Coordination |
| 2017 | 0.632 | 0.580 | −0.082 | Barely Coordination |
| 2018 | 0.645 | 0.598 | −0.072 | Barely Coordination |
| 2019 | 0.642 | 0.617 | −0.039 | Mild Coordination |
| 2020 | 0.547 | 0.637 | 0.164 | Mild Coordination |
| 2021 | 0.561 | 0.657 | 0.171 | Mild Coordination |
| 2022 | - | 0.678 | - | Mild Coordination |
| 2023 | - | 0.699 | - | Mild Coordination |
| 2024 | - | 0.721 | - | Moderate Coordination |
| 2025 | - | 0.744 | - | Moderate Coordination |
| 2026 | - | 0.767 | - | Moderate Coordination |
| 2027 | - | 0.792 | - | Moderate Coordination |
| 2028 | - | 0.817 | - | Benign Coordination |
| 2029 | - | 0.842 | - | Benign Coordination |
| 2030 | - | 0.869 | - | Benign Coordination |
| 2031 | - | 0.896 | - | Benign Coordination |
| Test Value | | C = 0.4914; $p$ = 0.8824 | | |

Table 8 shows, in the prediction model, that the C value is 0.4914 and the $p$ value is 0.8824, which reflects that the overall prediction accuracy is good (Table 5 is the accuracy criteria of the GM (1,1) model prediction), and the predicted data are representative. The relative error of prediction since 2009 is basically less than 0.1. However, the relative error is relatively large from 2020 to 2021, which is mainly related to the large changes in tourism in this period. According to the predicted value, the coupling and coordination state of tourism and ecological civilization in Guangdong Province has been gradually optimized and will improve over the next 10 years. The coupling and coordination degree will be greater than 0.8 from 2028, gradually entering a benign coordination stage. It is worth noting that the high-quality coordination stage has not been reached even in 2031. This demonstrates that the tourism and ecological civilization in Guangdong Province still needs to strengthen forward-looking thinking. On the premise of the normalization of epidemic prevention, how to scientifically and reasonably realize the value of ecological

products, innovative development, ecological priority and green development is worthy of significant consideration.

*5.3. Analysis of Gray Relation*

In this paper, the comprehensive score of the coupling coordination degree of the tourism industry and ecological civilization in Guangdong Province is taken as the reference sequence, and the index data of the tourism industry and ecological civilization system are taken as the alternative sequence to calculate the gray correlation coefficient between each alternative sequence and the reference sequence. The gray correlation score and ranking of the influencing factors of the comprehensive level of tourism industry and ecological civilization from 2005 to 2021 are sorted out. The frequency of the top 10 relevance indicators is extracted, as shown in Figure 7.

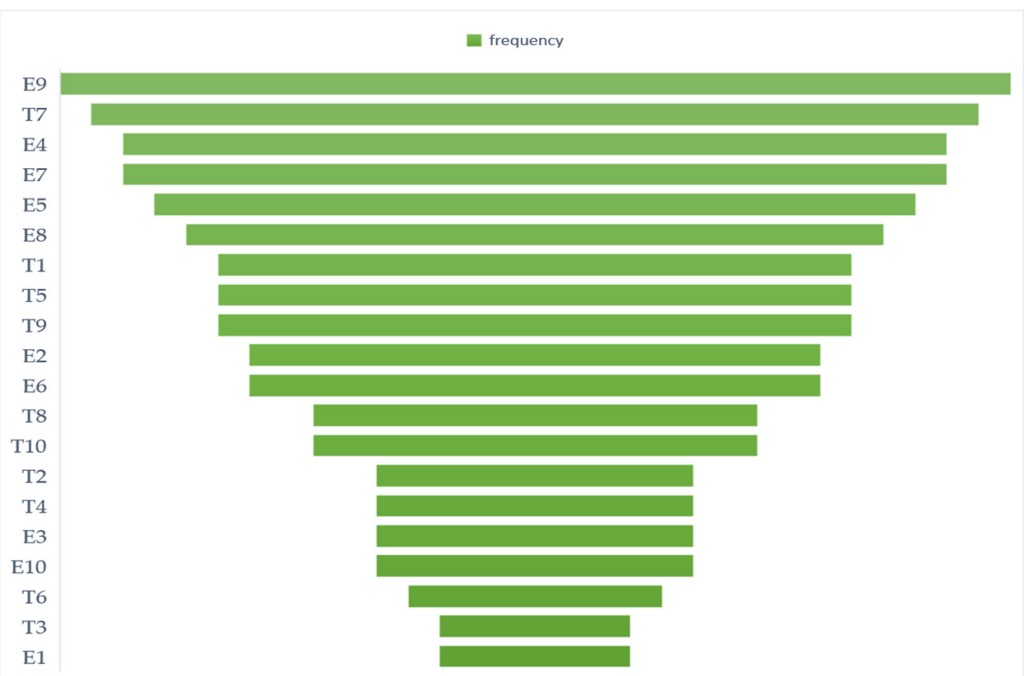

**Figure 7.** Frequency of top 10 indicators of gray coefficient between TI and EC (2005~2021).

From Figure 7, the rate of harmless garbage disposal (E9), the number of hotels (T7), the industrial wastewater (E4), and the per capita public green (E7) area are key factors to further enhance the coordinated development of tourism and ecological civilization in Guangdong Province. The forest coverage rate (E5), the rate of sewage centralized disposal (E8), the foreign exchange earnings (T1), the ratio of tourism revenue to GDP (T5) the employment in accommodation and catering (T9), the industrial SO$_2$ emissions (E2), and the green coverage rate in built-up area (E6) are the next most crucial factors to consider. Most of these factors belong to the ecological civilization category, which demonstrates the significance of adhering to the principle of "ecological priority and scientific utilization" to a certain degree, and the necessity of keeping the ecological bottom line, strengthening the awareness of ecological civilization during the process of tourism advancement in Guangdong Province.

*5.4. Analysis of Obstacle Factors*

The coupled and coordinated development of tourism and ecological civilization not only needs the top-level design of the macro-strategic planning but also needs the decisionmakers to identify the internal obstacles in order to carry out targeted improvement. In this regard, this paper further diagnoses and analyzes the obstacle factors of the coupling

and coordination between the TI-EC system according to the obstacle degree model, as shown in Figure 8.

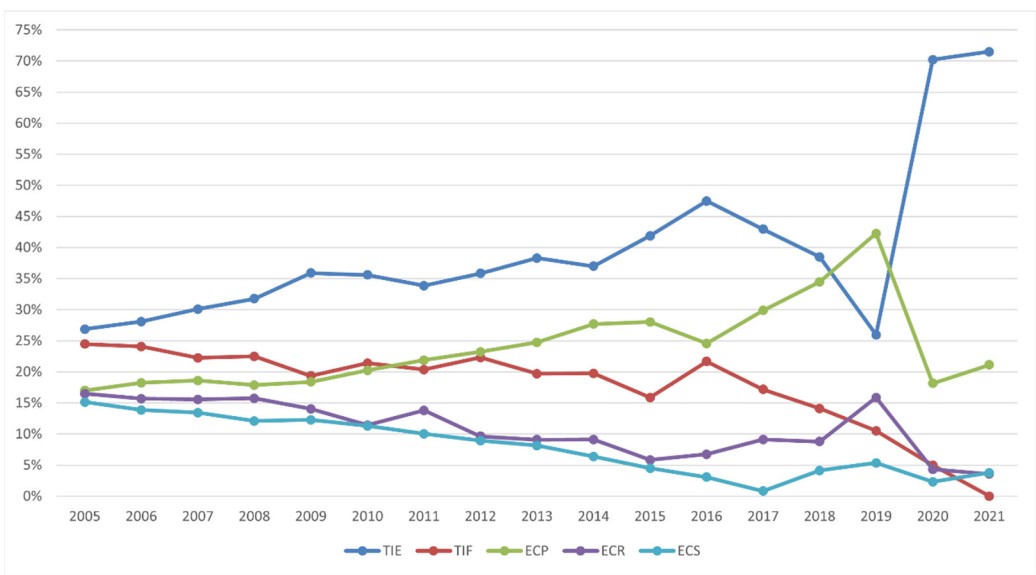

**Figure 8.** Obstacle factors trend of TI-EC system (2005~2021).

From the time dimension perspective, the average barrier degree of the first-class indicators from 2005 to 2021 is ranked from high to low as follows: Tourism Industry Effect (TIE, 39.52%), Ecological Civilization Pressure (ECP, 23.91%), Tourism Industry Foundation (TIF, 17.69%), Ecological Civilization Response (ECR, 10.88%), and Ecological Civilization Status (ECS, 7.99%). The difference between the results in each year during the sample period is not too obvious. We have further analyzed the secondary indicators and listed the top five obstacles over the years, as shown in Table 9.

Table 9 shows that during the study period, the second-level indicators ranking in the top five in barrier degree are relatively stable. From the perspective of obstacle intensity, domestic tourism earnings (T2), inbound tourists (T3), domestic tourists (T4), ratio of tourism revenue to GDP (T5), number of hotels (T7), A-level and above tourist attractions (T8), employment in accommodation and catering (T9), industrial $SO_2$ emissions (E2), industrial waste gas emissions (E1), industrial solid wastes produced (E3), and comprehensive utilization rate of industrial solid waste (E10) are the main factors hindering the coordinated development of cultural tourism and ecological civilization in Guangdong Province. According to the data in 2020 and 2021, domestic tourism earnings (T2), inbound tourism (T3), industrial waste gas emissions (E1), domestic tourists (T4), and ratio of tourism revenue to GDP (T5) remain basically stable, and the barriers are more than 10%, indicating that these factors are important hindering factors for tourism and ecological civilization coordination development in Guangdong Province under the epidemic environment.

Furthermore, to obtain a more detailed classification of these factors, the gray relation and obstacle degree of each secondary indicator have been analyzed synthetically and comprehensively based on the four-quadrant diagram, as shown in Figure 9.

**Table 9.** Main obstacle degree of coupling coordination within secondary index (2005~2021).

| | | 1 | 2 | 3 | 4 | 5 |
|---|---|---|---|---|---|---|
| 2005 | Obstacle | T2 | E2 | T4 | E8 | T8 |
| | Proportion | 10.64% | 10.71% | 8.38% | 8.22% | 8.15% |
| 2006 | Obstacle | T2 | E2 | E8 | T4 | T8 |
| | Proportion | 11.24% | 11.20% | 8.79% | 8.74% | 8.64% |
| 2007 | Obstacle | T2 | E2 | E8 | T7 | T4 |
| | Proportion | 10.52% | 10.07% | 9.01% | 8.11% | 8.06% |
| 2008 | Obstacle | T2 | E2 | E8 | T7 | T4 |
| | Proportion | 10.89% | 9.85% | 8.77% | 8.56% | 8.19% |
| 2009 | Obstacle | T2 | E2 | T4 | T8 | E8 |
| | Proportion | 12.32% | 10.52% | 9.05% | 8.44% | 7.16% |
| 2010 | Obstacle | T2 | E2 | T4 | T8 | T9 |
| | Proportion | 12.61% | 11.06% | 9.02% | 8.83% | 6.15% |
| 2011 | Obstacle | T2 | E2 | T4 | T8 | T9 |
| | Proportion | 12.16% | 9.48% | 8.63% | 8.53% | 5.68% |
| 2012 | Obstacle | T2 | E2 | T8 | T4 | T9 |
| | Proportion | 13.10% | 10.48% | 9.64% | 9.2% | 6.03% |
| 2013 | Obstacle | T2 | E2 | T8 | T4 | T3 |
| | Proportion | 13.66% | 11.44% | 10.39% | 9.46% | 7.4% |
| 2014 | Obstacle | T2 | E2 | T8 | T4 | T3 |
| | Proportion | 12.95% | 11.9% | 9.76% | 9.19% | 8.22% |
| 2015 | Obstacle | T3 | E2 | T2 | T8 | T4 |
| | Proportion | 15.32% | 12.28% | 12.11% | 10.04% | 8.58% |
| 2016 | Obstacle | T3 | E1 | T4 | T2 | T8 |
| | Proportion | 18.84% | 13.43% | 12% | 11.78% | 11.38% |
| 2017 | Obstacle | T3 | E1 | T8 | T2 | T4 |
| | Proportion | 22.71% | 17.87% | 12.4% | 9.69% | 6.75% |
| 2018 | Obstacle | T3 | E1 | T8 | E3 | E10 |
| | Proportion | 26.87% | 24.16% | 12.01% | 8.52% | 6.69% |
| 2019 | Obstacle | T3 | E1 | E3 | E10 | T8 |
| | Proportion | 25.97% | 25.68% | 15.91% | 15.15% | 9.44% |
| 2020 | Obstacle | T2 | T3 | E1 | T4 | T5 |
| | Proportion | 16.01% | 14.96% | 13.45% | 11.46% | 10.29% |
| 2021 | Obstacle | T3 | T2 | E1 | T4 | T5 |
| | Proportion | 15.93% | 15.53% | 15.42% | 10.85% | 10.71% |

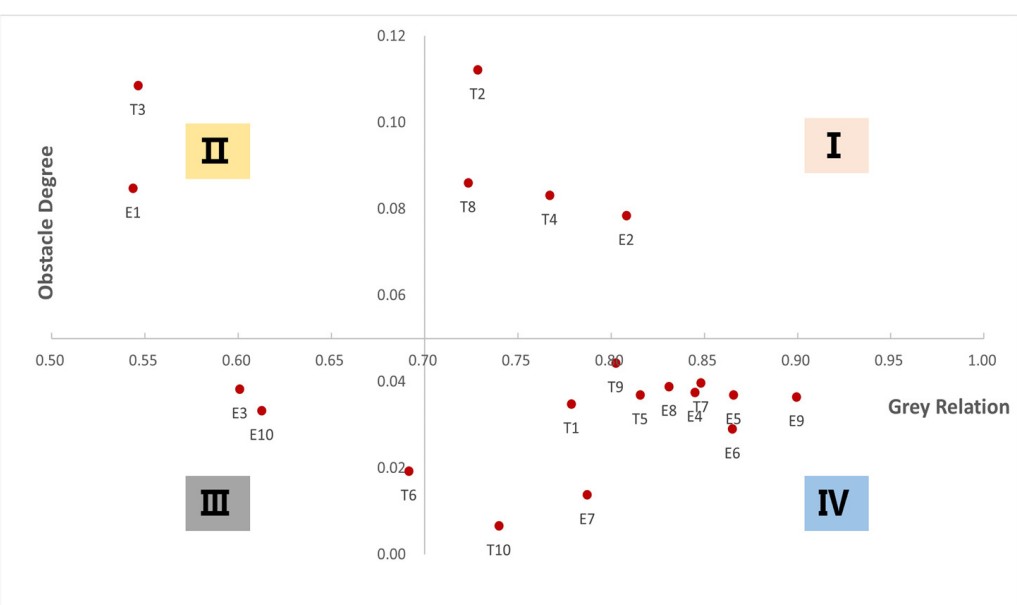

**Figure 9.** Four-quadrant diagram of gray relation and obstacle degree.

From Figure 9, all indicators have been divided into four categories. The first category is high-gray relation and high-obstacle, which is located in the upper right region. It is closely related to the coordinated development of tourism and ecological civilization in Guangdong Province but hinders the coordinated development of the two at this stage. Thus, it is a breakthrough to promote the coordinated development of tourism and ecological civilization. The second category is low-gray relation and high-obstacle, which is located in the upper left region. The third category is low-gray relation and low-obstacle, which is located in the lower left region. The fourth category is high-gray relation and low-obstacle, which is located in the lower right region. These indicators are the core supporting force for the coordinated development of tourism and ecological civilization.

Most indicators are in the high-gray relation region, which indicates the effectiveness of the indicators selected in this paper. Most of the indicators are within the region of high-correlation and high-barrier, and high-correlation and low-barrier zones, which reflects that there are still some dilemmas in the coupling and coordinated development of tourism and ecological civilization in Guangdong Province at this stage. In addition, with the extensive and far-reaching impact of the epidemic, the uncertainty of the global tourism industry has increased significantly, and the recovery of international tourism may have a certain lag period. Since the domestic tourism earnings (T2), domestic tourists (T2), A-level and above tourist attractions (T8), and industrial $SO_2$ emissions (E2) belong to the high-gray relation and high-obstacle factors, it is more urgent to seek progress while maintaining stability, to expand domestic tourism demand, to prosper and develop mass tourism, and to innovatively promote global tourism.

## 6. Conclusions

### 6.1. Findings

With the concerns of sustainable development, the harmonious coexistence between human and nature and how to better promote tourism development in the perspective of ecological civilization have become new issues. Nearly all countries are exploring new feasible development models. Starting from the strategic background of ecological civilization and tourism economic development, this paper takes data samples in Guangdong Province from 2005 to 2021 as an example for, respectively, constructing the tourism and the ecological civilization comprehensive evaluation index system. The comprehensive evaluation index of the tourism industry and ecological civilization, coupling coordination

degree, gray correlation, and obstacle degree factors have been measured and predicted. Some of the main findings are as follows:

1. The tourism industry of Guangdong Province is closely related to ecological civilization. Both show a slow rising trend and low starting point fluctuation. The average values of the comprehensive development of tourism and ecological civilization in Guangdong Province from 2005 to 2021 are 0.49 and 0.61, respectively, which are relatively close, indicating that they are closely related and have a certain integration basis. On the time scale, tourism lags behind ecological civilization, and there is a trend of catching up in 2019. However, affected by the epidemic, tourism has declined significantly in 2020. Although it shows a certain growth trend in 2021, it still lags behind ecological civilization again.

At the same time, based on the GM (1,1) model, the comprehensive development of the tourism industry and ecological civilization are predicted and analyzed, and both continue to show a growth trend in next decade.

2. The coordinated development of tourism and ecological civilization in Guangdong province still needs to be strengthened. From 2005 to 2021, there is an overall upward trend, reaching a peak in 2018. The overall evolution trend is "moderate disorder (2005–2008)—mild imbalance (2009–2011)—barely coordination (2012–2015)—mild coordination (2016–2019)". During the study period, it only reaches the mild coordination stage, and it falls back to the stage of barely coordination again in 2020 and 2021 with the impact of the epidemic and the other factors.

In the next decade, the coupling and coordination degree of tourism and ecological civilization in Guangdong Province continues to increase but keeps a mild coordination for two years. It will enter a moderate coordination stage in 2024, and in will enter a benign coordination state from 2028. However, it will still be in the benign coordination state in 2031 and not yet reach the stage of high-quality coordination. Overall, the coordinated development of tourism and ecological civilization in Guangdong Province presents upward trends while maintaining stability.

3. From the gray relation analysis perspective, each indicator has some difference in the coordinated development of tourism industry and ecological civilization in Guangdong Province from 2005 to 2021. Factors with a high degree of gray relation are derived from ecological civilization, such as the rate of harmless garbage disposal, the number of hotels, the industrial wastewater, and the per capita public green area. Thus, it is necessary to strictly protect the ecology and adhere to the path of ecological priority as well as green and sustainable development, which are beneficial for the coordinated development of the tourism industry and ecological civilization. Of course, it is necessary to pay attention to the impacts of tourism and ecological civilization at the same time.

4. Furthermore, according to the four-quadrant diagram synthetical analysis of gray relation and obstacle degree, the domestic tourism earnings (T2), domestic tourists (T2), A-level and above tourist attractions (T8), and industrial $SO_2$ emissions (E2) are closely related to the coordinated development of tourism and ecological civilization but hinders the two-coordinated development at this stage. It is the main breakthrough to promote the coordinated development of tourism and ecological civilization in Guangdong Province. Therefore, Guangdong Province needs to develop some tourism packages and use VR and cloud tourism technologies to achieve tourism "return". At the same time, measures such as resource conservation and emission reduction to optimize the ecological environment are also necessary implemented.

*6.2. Theoretical and Managerial Implications*

This paper contributes to tourism and ecological civilization literature by identifying the coupling coordination mechanism of the tourism and ecological civilization system with some comprehensive evaluation and prediction analysis. This paper provides several theoretical contributions. For example, this paper first constructs a coupling mechanism framework of a tourism ecological civilization system according to the connotation of tourism and ecological civilization system. Secondly, this paper also identifies and evaluates

the related factors and major obstacles in the TI-EC system. This may inspire more study to explore the impact of coordination and even to examine the results in different conditions, which may illuminate some new research issues.

This paper also provides some practical significance for the relevant management departments of tourism and ecological civilization. This paper forecasts the comprehensive development level of tourism and ecological civilization system in the next decade, which is conducive to understanding the development trend of tourism and ecological civilization and balancing the resources distribution. At the same time, this paper extracts, ranks, and analyzes the trend of the related factors and obstacle factors of the TI-EC system.

In terms of time sequence, the coupling and coordination degree of tourism and ecological civilization in Guangdong Province shows an overall upward trend, but the growth rate is relatively slow, and the high-quality coordination stage is still not reached in 2031. From the practice implications, this paper also makes the following suggestions:

1. Accelerate institutional innovation and create a coordinated environment for the development of tourism and ecological civilization. Overall, the coupling coordination degree between the tourism industry and ecological civilization in Guangdong Province is relatively low. High-quality development may require a cycle and bring some challenges to the initial development method and development planning [50]. Therefore, provincial, municipal, and local governments at all levels can contribute to planning and coordinating, improve the institutional system of tourism industry and ecological civilization, and build the incentive policies and restraint mechanisms of price, finance, taxation, and finance. It will be conducive to the mutual penetration of the two systems. Considering the infrastructure, public services, tourism route design and other aspects, while deepening the mining of tourism industry resources, the government should also pay attention to ecological environment protection, to emphasize development in protection and protection in development. In additional, creating a favorable social and cultural atmosphere to ensure the synchronous and coordinated development of the tourism industry and ecological civilization is also worth trying.

2. Adhere to the principle of adjusting measures to local conditions and expanding domestic tourism demand. This paper finds that the domestic tourism earnings, domestic tourists, A-level and above tourist attractions, and industrial $SO_2$ emissions are high gray relation and high obstacle factors for the coordinated development of tourism and ecological civilization. In this regard, Guangdong Province needs to highlight regional characteristics, actively develop new tourism formats, and meet the diversified tourism needs of multiple groups at all levels. At the same time, efforts can be made to launch a few core tourist attractions, deepen the development of smart tourism, and make full use of digital technology to strengthen tourism publicity and marketing planning. Eventually, it may help to broaden the tourist market, especially domestic tourism demand, and to accelerate the development of mass tourism.

3. Attach importance to the ecological civilization and promote the eco-tourism model. The overall level of ecological civilization is better than that of tourism at this stage. However, in the gray relation analysis, we found that most of the high correlation factors for the coordinated development of tourism and ecological civilization come from ecological civilization factors. In this regard, in the process of tourism advancement, we should be highlighting the systematic concept, strictly abiding by the bottom line of environmental protection, and striving to build several eco-tourism industry brands for Guangdong Province. These can enrich the connotation of tourism and ecological civilization and contribute to the integration and coordinated development of the two.

### 6.3. Limitations and Future Research

The tourism industry (TI) and ecological civilization (EC) have transformed into important real-time topics for regional economic development and urban construction. Influenced by the epidemic, more attention has been paid to how to achieve the return of the tourism industry in an innovative way and increase tourism attraction. Taking Guangdong



Province in China as an example, based on the main data from 2005 to 2021, this paper evaluated the coupling coordination in the tourism ecological civilization system and predicted the comprehensive development, coupling coordination degree and coupling coordination trend of the two systems in the next decade based on a GM (1,1) model. At the same time, based on gray relation analysis and the obstacle degree model, the paper analyzes the main factors and obstacle factors that affect the coordination of the two-coupling coordination. Some limitations still exist, which are outlined in the following.

Due to the difficulties in data collection and the incomplete data, this paper only analyzes the data from 2005 to 2021 and does not integrate the data from 2000 to 2004 for more in-depth analysis. At the same time, due to the complexity of the system and the differences in the study areas, the indicator systems are slightly different, which will also have a certain impact on the research results. In addition, this paper shows that the coupling and coordination of tourism and ecological civilization in Guangdong Province is on the rise as a whole and will continue to maintain this upward trend from 2022 to 2031. However, despite creating significant economic benefits, tourism also plays a key role in environmental degradation, sometimes even concealing the positive effects of tourism, especially in some developing countries [51]. The epidemic also increases the uncertainty of domestic and foreign tourism. In this regard, a more accurate prediction and analysis model can be used, such as the Markov model, to further explore how to promote the coordinated and sustainable development of tourism and ecological civilization within the ecological environment carrying capacity.

As pointed by some studies in the literature, it is also a hot spot in the research of the coupling and coordination of complex systems to integrate the spatial links from the geospatial perspective [18]. This paper has carried out a relatively in-depth multi-level analysis on the coupling and coordination degree of Guangdong Province and refined the main gray relation and obstacle factors from the time scale. However, the spatial correlation of regional development has not been considered. In this regard, in the future, it is also worthwhile to integrate the gravity model of new economic geography and regional economics to analyze the spatial interaction and depict the spatial connection structure of the coordinated development of tourism and ecological civilization in Guangdong Province at a more micro-scale. It may provide more enlightenment for management application to promoting the high-quality, coordinated, and orderly development between tourism and ecological civilization, which will be applicable to areas with the same industrial characteristics.

**Author Contributions:** Conceptualization, methodology, validation, formal analysis, writing—original draft preparation, D.Y.; writing—review and editing, visualization, supervision, D.Y. and G.J. All authors have read and agreed to the published version of the manuscript.

**Funding:** This research was funded by the 2021 Guangdong Provincial Department of Education Youth Innovation Talents Project "Integrated and innovative development of tourism industry and ecological civilization in Guangdong Province from the perspective of rural revitalization" (Project No.: 2021WQNCX092), which was initiated on 25 August 2021.

**Institutional Review Board Statement:** Not applicable.

**Informed Consent Statement:** Not applicable.

**Data Availability Statement:** Not applicable.

**Conflicts of Interest:** The authors declare no conflict of interest.

## Abbreviations

TI (Tourism Industry), EC (Ecological Civilization), TIF (Tourism Industry Foundation), TIE (Tourism Industry Effect), ECP (Ecological Civilization Pressure), ECS (Ecological Civilization Status), ECR (Ecological Civilization Response), UTI (Comprehensive Development of Tourism Industry), UEC (Comprehensive Development of Tourism Industry), GM (1,1) (Gray Model (1,1)).

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
