# Peer review of "Coupling Coordination Relationship between Tourism Industry and Ecological Civilization: A Case Study of Guangdong Province in China"

_sustainability, doi:10.3390/su15010092_

Round 1

Reviewer 1 Report

It's a good attempt, the detail comments for the improvement of this article are attached with.

Author Response

Sincerely appreciate for your review and valuable comments. I have revised them one by one. Please kindly see the attachment in detail. Sincerely appreciate for your review and guidance. 

Reviewer 2 Report

This study is a contribution to the scientific literature. It has a rigorous methodology and data collection process. The cited literature is up-to-date. The research findings and contributions are presented clearly. Therefore, I recommend this manuscript be accepted. 

Author Response

 Sincerely appreciate for your review and valuable comments. Because the paper has been revised and improved according to the received review comments. Please kindly see the attachment in detail. Sincerely appreciate for your review and guidance. 

Reviewer 3 Report

I draw the attention of the authors to a few things:

-the last paragraph in the Introduction is unnecessary

-from line 124 to 158 it is mentioned many times on the one hand-on the other hand – those sentences should be reformulated

-in the line 420- should be corrected So2 to SO2 (also in the table 7)

-paragraph 6.3 should move after conclusion

-I miss the connection and comparison with some other examples in the world, whether they exist and what are the results in other destinations

Author Response

(The authors gave the same response as above.)

Round 2

Reviewer 1 Report

Thank you for addressing the comments efficiently in the revised manuscript.